# Poor Sensitivity of Fecal Gluten Immunogenic Peptides and Serum Antibodies to Detect Duodenal Mucosal Damage in Celiac Disease Monitoring

**DOI:** 10.3390/nu13010098

**Published:** 2020-12-30

**Authors:** Emilio J. Laserna-Mendieta, María José Casanova, Ángel Arias, Laura Arias-González, Pedro Majano, Luis Alberto Mate, Carlos Humberto Gordillo-Vélez, Mirella Jiménez, Teresa Angueira, Emilia Tébar-Romero, María Jesús Carrillo-Ramos, María Ángeles Tejero-Bustos, Javier P. Gisbert, Cecilio Santander, Alfredo J. Lucendo

**Affiliations:** 1Department of Gastroenterology, Hospital General de Tomelloso, 13700 Tomelloso, Spain; laura.arias.gonzalez@gmail.com (L.A.-G.); teresa.angueira@gmail.com (T.A.); emiliatebar@hotmail.com (E.T.-R.); mjcr34@hotmail.com (M.J.C.-R.); angelestejero@hotmail.com (M.Á.T.-B.); 2Instituto de Investigación Sanitaria Princesa, 28006 Madrid, Spain; angel_arias_arias81@hotmail.com (Á.A.); pmajano@gmail.com (P.M.); cahugove@hotmail.com (C.H.G.-V.); 3Clinical Laboratory, Hospital Universitario La Princesa, 28006 Madrid, Spain; 4Department of Gastroenterology, Hospital Universitario de La Princesa, Instituto de Investigación Sanitaria Princesa (IIS-IP), 28006 Madrid, Spain; mjcasanova.g@gmail.com (M.J.C.); mirella_altea@hotmail.com (M.J.); javier.p.gisbert@gmail.com (J.P.G.); cecilio.santander@salud.madrid.org (C.S.); 5Enfermedades Inflamatorias Esófago-Gastro-Intestinales, Universidad Autónoma de Madrid (UAM), 28049 Madrid, Spain; 6Centro de Investigación Biomédica en Red de Enfermedades Hepáticas y Digestivas (CIBEREHD), 28222 Madrid, Spain; 7Research Unit, Hospital General Mancha Centro, 13600 Alcázar de San Juan, Spain; 8Pathology Department, Hospital General Mancha Centro, 13600 Alcázar de San Juan, Spain; lmatmcmar@gmail.com; 9Pathology Department, Hospital Universitario La Princesa, 28006 Madrid, Spain

**Keywords:** celiac disease, gluten-free diet monitoring, Marsh–Oberhüber type, gluten immunogenic peptides, anti-tissue transglutaminase antibodies, diagnostic accuracy

## Abstract

A lifelong gluten-free diet (GFD) is the only current treatment for celiac disease (CD), but strict compliance is complicated. Duodenal biopsies are the “gold standard” method for diagnosing CD, but they are not generally recommended for disease monitoring. We evaluated the sensitivity and specificity of fecal gluten immunogenic peptides (GIPs) to detect duodenal lesions in CD patients on a GFD and compared them with serum anti-tissue transglutaminase (tTG) IgA antibodies. A prospective study was conducted at two tertiary centers in Spain on a consecutive series of adolescents and adults with CD who maintained a long-lasting GFD. Adherence to a GFD and health-related quality of life were scored with validated questionnaires. Mucosal damage graded according to the Marsh–Oberhüber classification (Marsh 1/2/3) was used as the reference standard. Of the 97 patients included, 27 presented duodenal mucosal damage and 70 had normal biopsies (Marsh 0). The sensitivity (33%) and specificity (81%) of GIPs were similar to those provided by the two assays used to measure anti-tTG antibodies. Scores in questionnaires showed no association with GIP, but an association between GIPs and patients’ self-reported gluten consumption was found (*p* = 0.003). GIP displayed low sensitivity but acceptable specificity for the detection of mucosal damage in CD.

## 1. Introduction

Celiac disease (CD) is a chronic immune-mediated enteropathy, triggered and maintained by gluten consumption in genetically predisposed individuals, and characterized by damage of variable intensity in their duodenal mucosa [1]. A lifelong adherence to a gluten-free diet (GFD) is currently the only therapy for these patients, which usually leads to complete mucosal recovery. However, strict compliance to a GFD can be compromised by unnoticed gluten consumption, cross-contamination, or social pressure when eating out and unspecific or even absent symptomatology after transgressions [2]. As a consequence, 30% to 60% of patients with CD are exposed to gluten despite their best efforts [3,4,5].

Continuous gluten exposure causes permanent damage in the duodenal mucosa of celiac patients and increases the risk of bone and endocrinological diseases [6] or intestinal lymphoma [7]; lack of adherence to a GFD also reduces health-related quality of life (HRQoL) [8]. Monitoring sustained adherence to a GFD in celiac patients is, therefore, essential. This is mainly undertaken by surveying symptoms, administering nutritional questionnaires, and using non-invasive serological markers. The assessment of duodenal biopsies, which is considered the gold standard, is rarely performed during the follow-up of CD due to its invasiveness, relative risk, and high cost [9]. Persistence of duodenal mucosal lesions is more commonly found in adults compared to children and in those with a shorter time on a GFD [10,11].

There is a general consensus about the limitations of both patient-reported questionnaires and serological tests in detecting gluten transgressions and persistence of mucosal damage during the follow-up of CD [12,13]. Anti-tissue transglutaminase (tTG) and anti-endomysium antibodies are highly sensitive markers for the diagnosis of CD, but a meta-analysis showed their sensitivity in identifying villous atrophy fell below 50% once patients were on a GFD [14]. Therefore, celiac patients with negative antibody levels during follow-up could have undetected duodenal damage for years.

The detection in urine and fecal samples of gluten immunogenic peptides (GIPs), responsible for most of the immune-toxic reactions mediated by gluten in CD patients [15], has been proposed as being useful to check GFD compliance [16]. A manufacturer-sponsored study showed that 71% of the patients with Marsh type 2/3 lesions in duodenal biopsies showed negative serum antibodies, while positive urine GIPs were detected in all of the patients [17]. GIP measurement in stools, where they are present for 2–4 days after gluten ingestion, has been used to monitor adherence to GFD [18,19,20]. However, to date, no independent studies have evaluated the correlation between GIP detection in feces and changes in the duodenal mucosal histology in a real-life clinical setting.

In this study, our aim was to ascertain the utility of measuring GIPs with a qualitative lateral-flow immunochromatography assay in a single stool sample to detect duodenal mucosal lesions. To that end, we recruited adolescent and adult patients with CD in a clinical setting in two Spanish tertiary hospitals and assessed whether performing fecal GIP during CD monitoring is superior to serum tTG IgA antibodies’ measurement, as an index test, in detecting mucosal lesions by comparing them to the clinical reference standard: histopathological analysis of duodenal biopsies. In addition, HRQoL and patient-declared adherence to a GFD were measured to check for relationships between histological and laboratory results.

## 2. Subjects and Methods

### 2.1. Study Design and Reporting

The protocol was designed as a prospective observational study to evaluate the diagnostic accuracy of laboratory tests. The whole manuscript has been redacted to fulfill the essential items for the Standards for Reporting of Diagnostic Accuracy Studies, STARD 2015 [21].

### 2.2. Study Population

A consecutive series of adolescents and adult patients (age ≥ 14 years) with an established diagnosis of CD, more than one year on a GFD, and who attended either the Gastroenterology out-patient clinic at Hospital General de Tomelloso (HGT, Tomelloso, Spain) or Hospital Universitario de La Princesa (HUP, Madrid, Spain) between May 2018 and July 2020 were invited to participate in this study. Pregnant women, patients with other concomitant severe diseases, those with wheat allergy, non-celiac gluten sensitivity, and known refractory CD were not invited to participate. No prior calculation of sample size was performed. Recruited patients were given an appointment for an upper endoscopy and instructed to collect a fecal sample the day before the appointment. They were instructed to keep the sample at 4 °C until it was brought to the hospital.

### 2.3. Upper Endoscopy and Duodenal Biopsies

An esophago-gastro-duodenoscopy (EGD) under sedation with propofol was performed according to clinical practice. As per protocol, five biopsies from the second portion of the duodenum and two more from the duodenal bulb were obtained from each patient to investigate the persistence of mucosal lesions [22]. No reported adverse event occurred in any of the patients during the EGD.

The paraffin-embedded samples were cut, stained with hematoxylin–eosin, and studied under a light microscope by a single expert pathologist, blind to clinical and laboratory data, at each institution. Only representative and carefully orientated mucosal sections were included in the histopathological analysis. Quantification of the intraepithelial lymphocytes was performed through immunohistochemical examinations and using CD3 monoclonal antibodies for doubtful cases (Roche Diagnostics, Penzberg, Germany). The mucosal specimens were graded independently in accordance with the Marsh–Oberhüber classification: infiltration of intraepithelial lymphocytes >25% (type 1), crypt hyperplasia (type 2), and villous atrophy (type 3) [23,24]. This classification is widely accepted as a reference standard for CD assessment in clinical practice [22]. Presence of mucosal damage was considered when Marsh types 1, 2 or 3 were reported by the pathologists.

### 2.4. Measurement of GIPs

Patient stool samples were frozen at −40 °C as soon as they were delivered to medical staff, before the EGD. For GIP measurement, the samples were defrosted for 4–5 h and later homogenized. GIP detection was performed for all samples at HGT using iVYCHECK GIP-Stool (Biomedal, Sevilla, Spain) and following the manufacturer instructions. This assay is based on lateral-flow immunochromatography and provides a qualitative result. Interpretation was visually performed after 30 min of sample pouring. The presence of both green and red lines was considered GIP positive while the presence of only a green band was interpreted as GIP negative. Samples with the appearance of a green line at 30 min but not at 10 min were considered weak positives. No indeterminate results (presence of red band in absence of green band) were obtained. The limit of detection of the test was 0.15 µg GIP/g of stool, so this cutoff point represents the GIP concentration above which the test was positive.

GIP tests were performed in batches of several samples (normally 4 to 6) without knowing the reference standard results.

### 2.5. Measurement of Anti-tTG IgA

Blood from patients was extracted just before EGD for serum anti-tTG IgA determination. In addition, serum IgA levels were also measured to detect IgA deficiency, which was confirmed when the IgA concentration was lower than 70 mg/dL.

In HGT, anti-tTG IgA was measured with Liaison tTG IgA assay (DiaSorin, Saluggia, Italy) and manufacturer reference values were 0–8 UA/mL (lower limit of detection 0.2 UA/mL). Therefore, the test was interpreted as positive for anti-tTG concentrations above 8 UA/mL in the absence of IgA deficiency.

In HUP, anti-tTG IgA was measured with Quanta Flash IgA tTG assay (Werfen Diagnostics, Barcelona, Spain) and manufacturer reference values were 0–20 UA/mL (lower limit of detection 1.9 UA/mL). Consequently, the test was interpreted as positive for anti-tTG concentrations above 20 UA/mL in the absence of IgA deficiency.

At both sites, blood samples were processed in the corresponding auto-analyzers the day after the EGD, so the anti-tTG IgA result was completed for each patient before reference standard results were available. No indeterminate results were obtained from any sample, and patients with IgA deficiency were excluded from the final analyses.

### 2.6. Gluten-Free-Diet Adherence and HRQoL Questionnaires

On the same day and just prior to the EGD exam, patients were asked to complete two questionnaires previously adapted and validated linguistically and culturally in Spanish.

HRQoL was assessed using a celiac-specific questionnaire. This is a self-administered questionnaire, with 20 items across four clinically relevant subscales and answered using a Likert scale. The overall score is expressed on a scale of 20 to 100 points, with higher scores indicating better health and lower ones indicating higher stress levels due to CD [25].

The Spanish translation of the Celiac Dietary Adherence Test (CDAT) was used to evaluate adherence to the GFD; this consists of 7 items with a five-point scale for each item (7 to 35 points). Higher scores reflect poor adherence to GFD; with the last item asking about self-conscious gluten ingestion [26].

### 2.7. Statistical Analyses

Mean and standard deviations were reported for continuous variables and proportions for categorical data. The normality of continuous variables was checked using the Kolmogorov–Smirnov test. The Student t-test was employed for normal-distributed variables, while the U-Mann–Whitney test was used when normality was absent. For categorical data, contingency tables were produced and analyzed by chi-square or Fisher exact tests.

Sensitivity, specificity, positive predictive value (PPV), negative predictive value (NPV), and their 95% confidence intervals (95% CI) were calculated for GIP and tTG performance compared to the gold standard, which was the presence of any mucosal damage in the duodenum according to Marsh–Oberhüber classification. The percentage of agreement between methods was also calculated. The Kappa statistic was used for concordance between GIPs and mucosal lesions, as both were interpreted qualitatively. In addition, quantitative tTG values were employed to calculate the area under the receiver operating characteristic curve (AUC-ROC) and to estimate the optimal cutoff value by using the Youden index.

All statistical analyses were carried out using PASW version 18.0 (SPSS Inc., Chicago, IL, USA), GraphPad Prism version 5.0 (GraphPad Software, San Diego, CA, USA), and Epidat v4.2 (Servicio Gallego de Salud, Santiago de Compostela, Spain). Statistical significance was considered when *p* < 0.05.

## 3. Results

### 3.1. Demographic and Clinical Characteristics of Patients

Ninety-eight patients underwent EGD and provided fecal and serum samples (Figure 1). One patient from HGT was excluded due to a problem with biopsy sample processing at the Pathology Department. Table 1 shows the characteristics of the 97 patients finally included in the study. A female:male ratio of 3:1 was found, in agreement with the reported CD sex distribution. Mean age was close to 40 years old, and patients were on a GFD for a mean of 105 months, that is, almost 9 years. Most patients (62%) presented villous atrophy (Marsh 3) at the point of diagnosis.

Characteristics of patients recruited at the two hospitals were similar (Table 1), with no differences detected for most variables. However, serum anti-tTG was most frequently positive at HUP (*p* = 0.001), and more patients at HGT declared self-reported gluten consumption in the last month (*p* = 0.033).

No differences were observed in the age of the patients with complete mucosal recovery (Marsh type 0) compared to those with Marsh 1/2/3 (mean and SD being 38.9 ± 18.0 vs. 41.9 ± 18.1 years; *p* = 0.466). However, complete mucosal recovery was associated with longer duration of the GFD rather than presence of any mucosal damage (114.9 ± 96.7 vs. 78.9 ± 62.9 months; *p* = 0.035).

### 3.2. Performance of GIP Detection in Stool Samples to Identify Mucosal Damage

GIPs were detected in 22.7% of the patients, with no differences between recruiting hospitals and with more than half of the samples (55%) being weak positives. No differences in sex, age, length on a GFD, and Marsh type at diagnosis were observed between patients with positive and negative GIP results.

Compared to duodenal histology as the gold standard, the sensitivity and specificity of GIPs to detect any mucosal damage were 33.3% (CI 95%, 18.6–52.2%) and 81.4% (CI 95%, 70.8–88.8%), respectively (Table 2). PPV and NPV were 40.9% (CI 95%, 23.3–61.3%) and 76% (CI 95%, 65.2–84.2%), respectively, with an agreement of 68% and a kappa of 0.157 (CI 95%, −0.053 to 0.366). GIPs were undetected in 4 of the 6 patients with the most severe mucosal damage (Marsh type 2/3) (Table 3).

### 3.3. Performance of Serum Anti-tTG IgA Antibodies to Detect Mucosal Damage

Anti-tTG IgA assays were different in the two recruiting hospitals, resulting in different concentrations and positivity rates. Therefore, both assays were analyzed separately.

Liaison anti-tTG assay showed no significant association with Marsh type (*p* = 0.074), and the sensitivity and specificity to detect any mucosal lesion were 11.8% (CI 95%, 3.3–34.3%) and 100% (CI 95%, 92–100%), respectively (Table 2). The only two patients with tTG concentration above the cutoff value (8 UA/mL) presented mucosal lesions, thus resulting in high specificity. The agreement was 75.4%, with a PPV and NPV of 100% (CI 95%, 34.2–100%) and 74.6% (CI 95%, 62.2–83.9%), respectively. The AUC-ROC was 0.549 (CI 95%, 0.380–0.718) (Appendix A), with an optimized cutoff concentration of 0.85 UA/mL, which, however, displayed limited sensitivity (64.7%) and specificity (50%).

Likewise, poor sensitivity (30%, CI 95%, 6.7–65.3%), moderate specificity (72.7%, CI 95%, 49.8–89.3%), and no association with Marsh type (*p* = 1.0) was observed for the Quanta Flash anti-tTG assay (Table 2). Only one out of three patients with Marsh 3 biopsies displayed tTG levels above the cutoff (20 UA/mL) (Table 3). The agreement was lower than for the Liaison assay (59.4%), providing also a lower PPV (33.3%, CI 95%, 13.5–61.6%) and NPV (69.6%, CI 95%, 58.6–78.7%). The AUC-ROC was similar to that calculated for the Liaison anti-tTG assay (0.564, CI 95%, 0.348–0.780) (Appendix A), with equally limited performance of the optimized cut-off concentration (4.0 UA/mL, sensitivity = 80% and specificity = 40.9%).

### 3.4. Association between Serum Antibodies and GIP Results

No differences were detected for quantitative anti-tTG serum values between GIP-positive and GIP-negative patients for any of the anti-tTG assays (*p* = 0.884 for Liaison assay and *p* = 0.755 for Quanta Flash assay) (Appendix A).

If serum anti-tTG results were interpreted as positive or negative according to the manufacturers’ recommended cutoff, a close-to-significance association was found between Liaison anti-tTG assay and stool GIPs (*p* = 0.050), as GIPs were detected in the two patients with positive tTG. Conversely, no relationship was observed between Quanta Flash anti-tTG assay and GIP detection (*p* = 1.0).

### 3.5. Patients’ Scores from Questionnaires and Relationship with Demographics, Mucosal Damage, and Laboratory Markers

HRQoL and adherence to GFD (CDAT score) inversely correlated (Pearson’s r = −0.099 ± 0.032; *p* = 0.003) in the sense that a poorer QoL level was associated with a lower adherence to GFD (Appendix A).

However, no significant association was observed between HRQoL and CDAT scores when groups were compared for the following variables: age group (14–25 years, 26–55 years, >55 years), sex, length on a GFD (12–36 months, 37–120 months, >120 months), presence of mucosal damage (Marsh 0 vs. Marsh 1/2/3), and GIP detection. The only association found was patients with positive anti-tTG recruited at HUP showing worse adherence to a GFD (higher CDAT scores) than those with negative ones (14.6 ± 5.8 vs. 11.1 ± 3.2; *p* = 0.038).

An association between patient self-reported gluten consumption in the previous month (last question of CDAT questionnaire) and GIP detection was found (*p* = 0.003). GIPs were more frequently detected in patients who were aware of any gluten consumption in the last month (52.9%) than in those who believed that they had not ingested gluten at all (16.3%) (Table 4). This also showed that almost 60% of patients with positive GIPs were not aware of gluten transgressions.

## 4. Discussion

Patients with CD should be monitored to ensure the achievement of positive health outcomes and an appropriate reversion of the mucosal damage caused by gluten. However, monitoring strict compliance of a GFD is controversial in CD management [27]. Although there are several options to evaluate this compliance (presence of symptoms, HRQoL questionnaires, self-reported adherence to a GFD, nutritional counseling, dietary reports, serological assays, and endoscopy with duodenal biopsies), there is no consensus on a suitable methodology and frequency of follow-up in practice guidelines [9,28]. In our study, we have assessed the diagnostic accuracy of stool GIPs measured with a qualitative assay to detect mucosal damage in CD patients on a long-term GFD by comparing GIP results with the reference standard method. To our knowledge, this is the first study without GIP manufacturer funding addressing the capability of stool GIPs to detect mucosal damage during monitoring of CD patients in a real-life clinical setting. According to our results, GIPs had similar sensitivity and specificity to serum anti-tTG IgA for detecting any mucosal damage (Marsh type 1/2/3), with 22.7% of patients having gluten transgressions according to single stool GIP detection.

Persistence of mucosal damage is associated with non-responsive CD, which is mostly caused by gluten exposure and rarely (1–3% of patients) by refractory CD [29]. However, current consensus does not recommend endoscopy for routine monitoring of CD, only for complicated cases with persisting symptoms and laboratory-proved deficiencies in micronutrients [9]. Therefore, it is not rare that asymptomatic patients with negative serology have no second endoscopy after CD diagnosis for years, thus missing possible mucosal lesions [13]. Although recognized as the reference standard method, the specific role of endoscopic surveillance in CD follow-up still needs to be defined [27,30].

Severe mucosal damage (Marsh type 2/3) was found in 6.2% of our cohort, less than expected according to some previous publications [3,4,5,31], but in agreement with other studies reporting ratios of 4–6% after a long-term GFD [32,33]. Patients with mild mucosal damage (Marsh 1 or intraepithelial lymphocytosis) represented 21.6% of our cohort. We assumed that this was caused by persistent CD activity, although mild mucosal damage could be present in other conditions, such as *Helicobacter pylori* infection, parasitic infections (mainly, *Giardia lamblia*), small intestinal bacterial overgrowth, lymphocytic colitis, food allergies, or non-steroidal anti-inflammatory treatment [1]. Marsh 1 lesions are present in some CD patients who strictly adhere to a GFD [32] and even persist after GFD optimization [34], so the clinical significance and long-term impact of prolonged duodenal intraepithelial lymphocytosis needs to be investigated further.

As expected, serum anti-tTG IgA assays from different manufacturers varied in numerical values and positivity rates [35]. Despite serum anti-tTG usually falling below the positive threshold one-year after setting up a GFD [36], serology shows low sensitivity for detecting mucosal damage beyond CD diagnosis [14]. Therefore, negative serology does not necessarily indicate mucosal healing. Some alternatives were proposed to improve anti-tTG performance, including the use of undetectable levels as a marker of damage absence [31], employment of reference change values [37], or defining “compliance” in addition to “diagnostic” cutoffs [38]. Regarding the latter, optimized cutoff points according to the Youden index still showed poor diagnostic accuracy in the anti-tTG assays tested, with increases in sensitivity balanced by great decreases in specificity. In our study, the main advantage of Liaison anti-tTG assay was the absence of false positives, as the two patients with positive concentrations had mucosal damage, while positive results in Quanta Flash assay were associated with lower adherence to a GFD measured by the CDAT questionnaire. In contrast, no association was observed between serum anti-tTG IgA concentrations and GIP results as also reported in previous studies [20,39], although others did find agreement between the two tests [40].

A single GIP determination displayed positive results in 22 patients (22.7%) and was associated with self-reported gluten consumption. Initial studies assessing urine or stool GIPs provided higher rates of 29.8% [16] and 46.5% [17], with positivity increasing further to between 58% to 89% when several samples were taken from the same patient over a period [20,39]. In these latest two studies, an increased span of testing, however, led to detecting patients with positive GIPs but no duodenal damage. In our cohort, GIPs were also detected in 13 patients (13.4%) with no mucosal damage (Marsh type 0), with 5 of them reporting gluten consumption the month before EGD. A potential tolerance of small amounts of gluten [41] or delay in appearance of mucosal damage could explain this finding, which deserves further investigation. Anyway, although frequency and number of determinations for stool GIP testing have not been established for clinical routine, our results and others from previous publications indicate that the more tests performed, the more patients without mucosal damage will have positive GIPs, which could result in unnecessary increases in costs of CD follow-up.

It should be noted that stool GIPs reflect gluten consumption 2–4 days before the sample is provided, so its use as a surrogate marker for mucosal damage, as suggested in some manufacturer-funded studies, needs to be evaluated carefully. Manufacturer-funded research assessing this issue showed good agreement (80–100%) between persistence of villous atrophy (Marsh types 2/3) and detectable GIPs in either urine [17,39] or both stool and urine [19]. However, our study found negative GIPs in 4 out of 6 patients with Marsh type 2/3 using a single determination in stool, overall providing remarkably low sensitivity (33.3%) and PPV (40.9%). Therefore, it would be advisable to combine GIP with other tools. A recent study suggested combining fecal GIPs with a CDAT questionnaire as the best option for monitoring dietary compliance in celiac patients [42].

Among the limitations of our study, the low number of patients with villous atrophy prevented us from specific analyses for this group of patients. The long period of our cohort on a GFD was the likely cause of this, as reported in other cohorts with long lasting adherence. Marsh type 1 was interpreted as being caused by CD, but other potential causes of intraepithelial lymphocytosis should also be acknowledged. Knowing that the effectiveness of the diet was going to be evaluated could have led to better adherence by our patients, which is an intrinsic problem of GIP testing. Finally, patients with refractory CD were not included, nevertheless the potential impact of this rare condition on our results would be negligible given its extremely low prevalence.

## 5. Conclusions

To conclude, stool GIP testing could be useful in certain CD patients to identify gluten transgression leading to persistent mucosal damage, but its sensitivity and specificity in detecting histopathological lesions for a single measurement was not superior to that observed for serum anti-tTG IgA. Finding non-invasive surrogate biomarkers of persistent mucosal damage in CD is still a requirement, and the employment of different tools for CD monitoring, depending on the particular clinical situation of each patient, would be the most advisable approach.

## Figures and Tables

**Figure 1 nutrients-13-00098-f001:**
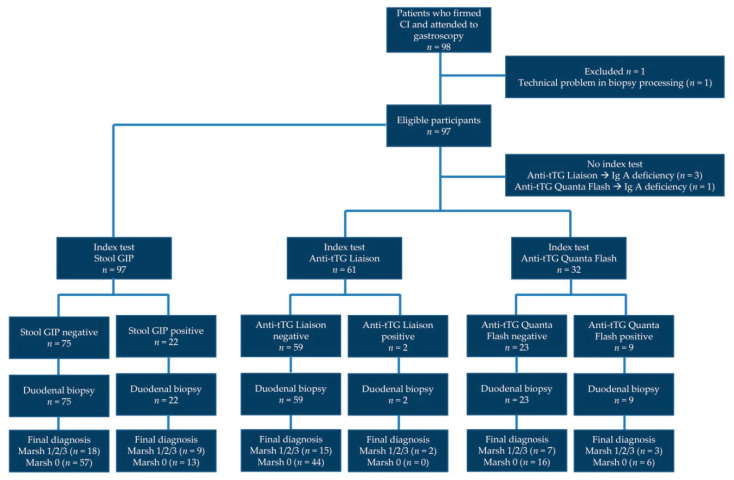
Flowchart of participants through the study.

**Table 1 nutrients-13-00098-t001:** Demographic and clinical characteristics of patients included in the study and statistical comparison between both recruiting hospitals.

		Total	HGT	HUP	*p*
Patients		97	64	33	
Sex	Female	75 (77.3%)	48 (75.0%)	27 (81.8%)	0.477
	Male	22 (22.7%)	16 (25.0%)	6 (18.2%)	
Age (years)		39.7 ± 18.0 (14–77)	38.0 ± 19.2 (14–75)	43.2 ± 15.1 (20–77)	0.180
Age (years) at diagnosis		31.0 ± 19.9 (0–72)	29.1 ± 21.1 (1–68)	34.6 ± 17.1 (0–72)	0.168
Months in GFD		104.9 ± 89.7 (12–362)	106.3 ± 84.4 (12–362)	102.1 ± 100.6 (12–318)	0.826
Marsh type at diagnosis (%)	Marsh 3	60 (61.9%)	38 (59.4%)	22 (66.7%)	0.313
	Marsh 2	9 (9.3%)	8 (12.5%)	1 (3.0%)	
	Marsh 1	11 (11.3%)	7 (10.9%)	4 (12.1%)	
	Unknown	17 (17.5%)	11 (17.2%)	6 (18.2%)	
Marsh type currently (%) ^a^	Marsh 3	3 (3.1%)	0	3 (9.1%)	0.695
	Marsh 2	3 (3.1%)	3 (4.7%)	0	
	Marsh 1	21 (21.6%)	14 (21.9%)	7 (21.2%)	
	Marsh 0	70 (72.2%)	47 (73.4%)	23 (69.7%)	
Positive fecal GIPs (%) ^b^		22 (22.7%) [54.5%]	15 (23.4%) [53.3%]	7 (21.2%) [57.1%]	0.804
Positive serum anti-tTG IgA (%) ^c^		11 (11.8%)	2 (3.3%)	9 (28.1%)	**0.001**
Health-related quality of life score ^d^		71.6 ± 12.1 (39–96)	71.0 ± 12.7 (39–96)	72.7 ± 10.9 (40–89)	0.520
CDAT score ^e^		12.2 ± 4.0 (7–27)	12.2 ± 3.9 (7–24)	12.1 ± 4.3 (7–27)	0.910
Self-reported gluten consumption (%) ^f^		17 (17.5%)	15 (23.4%)	2 (6.1%)	**0.033**

HGT: Hospital General de Tomelloso; HUP: Hospital Universitario de La Princesa; GFD: gluten-free diet; GIPs: gluten immunogenic peptides; tTG IgA: tissue transglutaminase immunoglobulin A; CDAT: Celiac Dietary Adherence Test. Significant differences between patients recruited in different hospitals are highlighted in bold. ^a^ Marsh 2 and Marsh 3 were grouped together for the statistical comparison between hospitals. ^b^ The percentage of total positives considered as weak positives (GIP band visible at 30 min but not at 10 min) is showed in brackets. ^c^ Three patients from HGT and one patient from HUP are not included as they presented IgA deficiency. ^d^ Higher scores indicate better quality of life. ^e^ Higher scores indicate worse adherence to a GFD. ^f^ Patient self-reported gluten consumption in the last month based on the response to the last question of adherence to GFD questionnaire.

**Table 2 nutrients-13-00098-t002:** Diagnostic performance of presence of gluten immunogenic peptides (GIPs) in stool and serum anti-tissue transglutaminase (tTG) IgA assays to detect mucosal damage in duodenal biopsies.

	Marsh0	Marsh1/2/3	Total	AUC	AG	SE	SP	PPV	NPV
GIP (iVYCHECK, Biomedal)									
Positive (>0.15 µg/g)	13	9	22	-	68.0%	33.3%	81.4%	40.9%	76.0%
Negative (<0.15 µg/g)	57	18	75						
Anti-tTG IgA (Liaison, Diasorin)									
Positive (>8 UA/mL)	0	2	2	0.549	75.4%	11.8%	100%	100%	74.6%
Negative (<8 UA/mL)	44	15	59						
Anti-tTG IgA (Quanta Flash, Werfen)									
Positive (>20 UA/mL)	6	3	9	0.564	59.4%	30.0%	72.7%	33.3%	69.6%
Negative (<20 UA/mL)	16	7	23						

AUC: area under the receiver operating characteristic curve; AG: agreement; SE: sensitivity; SP: specificity; PPV: positive predictive value; NPV: negative predictive value.

**Table 3 nutrients-13-00098-t003:** Demographic data, clinical characteristics, and laboratory results for patients with Marsh type 2/3 in the duodenal biopsy.

Biopsy	Centre	Sex	Age (y)	GFD Duration (m)	Marsh (dx)	GIPs	Anti-tTG IgA (UA/mL)	IgA (mg/dL)	HRQoL Score	CDAT Score	Gluten Intake
Marsh 2	HGT	Female	18	194	Marsh 3	Neg	2.0	305	80	17	Yes
			16	38		Pos	1.6	205	64	11	
		Male	19	77			14.2	329	58	17	
Marsh 3	HUP	Female	63	41	Marsh 3	Neg	169.9	258	62	19	No
			39	56			4.1	212	86	9	
			26	302	Unknown		<1.9	172	85	14	

HGT: Hospital General Tomelloso; HUP: Hospital Universitario de La Princesa; y: years; dx: when celiac disease was diagnosed. GFD: gluten-free diet; m: months; GIPs: gluten immunogenic peptides in stool; Neg: negative; Pos: positive; tTG IgA: tissue transglutaminase immunoglobulin A; HRQoL: health-related quality of life; CDAT: Celiac Dietary Adherence Test. Gluten intake was based on self-reported gluten consumption in the last question of the adherence to gluten-free diet questionnaire.

**Table 4 nutrients-13-00098-t004:** Relationship between detection of gluten immunogenic peptides (GIPs) in stool and patient self-reported gluten consumption in the last month before gastroscopy was performed.

	GIP Negative	GIP Positive	Total
Gluten consumption	8 (47.1%)	9 (52.9%)	17
No gluten consumption	67 (83.7%)	13 (16.3%)	80
	75	22	97

## Data Availability

The data presented in this study are available on request from the corresponding author. The data are not publicly available due to protection of patients’ confidential data.

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
