# Peer review of "Poor Sensitivity of Fecal Gluten Immunogenic Peptides and Serum Antibodies to Detect Duodenal Mucosal Damage in Celiac Disease Monitoring"

_nutrients, 2020, doi:10.3390/nu13010098_

Round 1

Reviewer 1 Report

Authors conducted a prospective study in celiac patients on long-lasting gluten-free diet to investigate the role of fecal gluten immunogenic peptides (GIP) to detect duodenal lesions

 Sensitivity (33%) and specificity (81%) of GIP were similar to those provided by anti-tTG antibodies tests, therefore GIP showed low sensitivity but acceptable specificity in disease monitoring. At date, the evaluation of adherence to a GFD during clinical follow-up is debated. Mucosal healing would be the ideal parameter to monitor gluten-free adherence, but invasiveness, relative risk, and cost limit biopsy sampling procedure. The potential role of fecal gluten immunogenic peptides (GIPs) as a non-invasive method of assessing dietary compliance in celiac patients suggests that they are a promising tool. The topic is interesting, and study was properly conducted. English language is accurate. Statistical analysis is adequately conducted. The Materials and methods section is appropriate, Ethics Statement is clearly detailed. The Results description is well organized. Authors elaborated figures and tables carefully.

I suggest to expand the Discussion section, maybe with the results of the most recent publications on the topic (see Porcelli et al. Ann Gastroenterol. 2020 Nov-Dec;33(6):631-637; Larussa et al. Minerva Gastroenterol Dietol. 2020 May 13; Porcelli et al. Minerva Gastroenterol Dietol. 2020 Sep;66(3):201-207).

Author Response

We thank the reviewer for his/her evaluation of our manuscript. The discussion section has been expanded as suggested by the reviewer (new text inserted is highlighted in yellow). New references to the two papers by Porcelli et al (references # 40 and 42) are included in the revised version of the manuscript.

Reviewer 2 Report

The present study was well designed and the results are very useful in daily clinical practice in the care of subjects with celiac disease.I recommend the authors to simplify Table 1, unifying the positive and negative results of the anti-transglutaminase antibodies derived from the two clinical centers. Consequently, Figure 1 should also be simplified. Table 3 should be modified by unifying the clinical and histological data of the six patients into only two homogeneous groups.The discussion should be shortened by focusing the discussion among the objectives described in the final part of the introduction.

Author Response

1. I recommend the authors to simplify Table 1, unifying the positive and negative results of the anti-transglutaminase antibodies derived from the two clinical centers.

We are grateful to this reviewer for such a detailed review of our manuscript. Regarding to Table 2 (we believe the reviewer is probably referring to Table 2 instead to 1 in the above comment), results for serum anti-tTG IgA were split according to the assay used in each of the recruiting hospitals. We appreciate the reviewers recommendation but, as results were obtained with different assays which were not interchangeable in terms of positivity thresholds, it would be not accurate to merge results from different techniques in a single table. It is generally accepted that results from anti-tTG IgA assays show relevant differences among them (please consider Li M et al, Am J Gastoenterol, 2009, 104:154-163; Naiyer AJ et al, J Clin Gastroenterol, 2009, 43:225-232 as examples of studies that observed significant differences among assays from different manufacturers). This fact is briefly explained in the discussion section (lines 311-312). As we also found significant differences in positivity rates between the two anti-tTG IgA assays (as shown in Table 1), we consider that describing and analyzing anti-tTG results for each particular assay separately is fully justified.

2. Consequently, Figure 1 should also be simplified.

For the reason explained in the previous point, we respectfully consider that it is more accurate to describe anti-tTG results separately for each particular assay.

3. Table 3 should be modified by unifying the clinical and histological data of the six patients into only two homogeneous groups.

Changes have been done in Table 3 in order to simplify it by grouping those variables that were common among patients. The column age (years) at diagnosis has been removed as this information was redundant with the duration of the gluten-free diet. However, we have maintained the particular information for each patient as we consider it could be interesting for readers looking for specific data from patients who presented severe mucosal damage in our study.

4. The discussion should be shortened by focusing the discussion among the objectives described in the final part of the introduction.

We appreciate this suggestion. However, Reviewer #1 requested to expand the discussion by quoting new references. Therefore, two new sentences were added to the revised manuscript. As there is no limit in the word count in Nutrients, we prefer to keep the discussion with the current content and extension. Anyway, we are fully open to remove some specific paragraphs of the discussion if required by Reviewer #2 or to rewrite any part that the reviewer considers could be out of the scope of the paper.